# The Current State of Neoadjuvant Therapy in Resectable Advanced Stage Melanoma

**DOI:** 10.3390/cancers15133344

**Published:** 2023-06-26

**Authors:** Omar Bushara, Jerica Tidwell, James R. Wester, John Miura

**Affiliations:** Department of Surgery, Hospital of the University of Pennsylvania, Philadelphia, PA 19104, USA; omar.bushara@pennmedicine.upenn.edu (O.B.); jerica.tidwell@pennmedicine.upenn.edu (J.T.); james.wester@pennmedicine.edu (J.R.W.)

**Keywords:** melanoma, neoadjuvant, immunotherapy, targeted therapy

## Abstract

**Simple Summary:**

The current standard of care for locally advanced melanoma is surgery first followed by systemic therapy. However, there is growing evidence that neoadjuvant therapy may be beneficial. The current literature supports that neoadjuvant therapy may downstage tumors and thus reduce the extent of needed surgery, allow for prognostication based on the initial response to therapy, and is associated with improved outcomes. The goal of this article is to review clinical trials of neoadjuvant therapy in locally advanced melanoma.

**Abstract:**

The advent of effective immunotherapy and targeted therapy has significantly improved outcomes in advanced-stage resectable melanoma. Currently, the mainstay of treatment of malignant melanoma is surgery followed by adjuvant systemic therapies. However, recent studies have shown a potential role for neoadjuvant therapy in the treatment of advanced-stage resectable melanoma. Mechanistically, neoadjuvant immunotherapy may yield a more robust response than adjuvant immunotherapy, as the primary tumor serves as an antigen in this setting rather than only micrometastatic disease after the index procedure. Additionally, targeted therapy has been shown to yield effective neoadjuvant cytoreduction, and oncolytic viruses may also increase the immunogenicity of primary tumors. Effective neoadjuvant therapy may serve to decrease tumor size and thus reduce the extent of required surgery and thus morbidity. It also allows for assessment of pathologic response, facilitating prognostication as well as tailoring future therapy. The current literature consistently supports that neoadjuvant therapy, even as little as one dose, is associated with improved outcomes and is well-tolerated. Some patients with a complete pathological response may even avoid surgery completely. These results challenge the current paradigm of a surgery-first approach and provide further evidence supporting neoadjuvant therapy in advanced-stage resectable melanoma. Further research into the optimal treatment schedule and dose timing is warranted, as is the continued investigation of novel therapies and combinations of therapies.

## 1. Introduction

Malignant melanoma is the fifth most common cancer in the US, with almost 100,000 new cases diagnosed in 2022 [1]. Additionally, it is the deadliest skin cancer, although the 5-year survival has significantly improved in recent times [2,3]. This improvement in outcomes has, in part, been due to the advent of effective systemic therapies in patients with American Joint Committee on Cancer Stage III and IV disease [4,5,6]. Current guidelines denote surgery followed by adjuvant therapy as the mainstay of treatment in those with resectable stage III/IV disease. However, more recent literature focusing on a neoadjuvant treatment sequence has challenged the conventional treatment paradigm and exposed the potential advantages of delivering these systemic agents before surgery [7,8]. Neoadjuvant therapy, particularly immunotherapy, may be beneficial by yielding a more potent immune response than adjuvant therapy, decreasing the size of lesions and thus decreasing the extent of required surgery, and allowing more frequent time points to assess response, progression, and recurrence over the course of treatment. Herein, we describe current evidence for neoadjuvant therapy in the treatment of advanced resectable melanoma and identify areas of potential future research. 

## 2. Rationale

Mechanistically, neoadjuvant immunotherapy may yield a more robust response than an adjuvant treatment sequence. Current immunotherapy involves checkpoint blockade of programmed cell death protein 1 (PD-1) and cytotoxic T-lymphocyte-associated protein 4 (CTLA-4). These receptors physiologically function to dampen the immune response. PD-1 is expressed primarily on cytotoxic CD8+ T lymphocytes and binds to programmed cell death ligand 1 (PD-L1) expressed on tumor cells, while CTLA-4 is expressed on regulatory T cells and other activated T cells and binds to B7 expressed on antigen-presenting cells. The binding of both receptors to these ligands reduces anti-tumor cytotoxic activity and contributes to T cell exhaustion [9,10,11,12,13,14]. PD-L1 is also highly expressed on dendritic cells, crucial in the anti-tumor response in melanoma [15], and the binding of PD-1/PD-L1 on CD8+ T cells and dendritic cells has been shown to also downregulate CD8+ T cell activity [16]. As such, anti-PD-1 and anti-CTLA-4 therapy primarily serve to improve the host immune response to primary malignancies [9]. More recently, relatlimab, a lymphocyte-activation gene 3 (LAG-3)–blocking antibody that functions to reduce LAG-3 mediated T cell inhibition, has been developed and used as an additional immunotherapy for melanoma [17,18].

Crucial to these therapies is the presence of a tumor against which an immune response may be directed. The primary tumor serves to be antigenic and yields a specific anti-tumor immune response [19]. The introduction of immunotherapy thus primes effector T cells, mitigates cytotoxic T cell exhaustion, and enhances the activity of T cells that have already infiltrated the tumor microenvironment [19,20,21]. These effects have an additive benefit in patients with “hot” tumors and may serve to turn “cold” tumors more immunogenic [22]. Comparatively, adjuvant immunotherapy is introduced at a stage where the primary antigen has been removed, and thus the anti-tumor immune response may not be as robust if directed at potentially only micrometastatic disease [20]. Plausibly, this would be a less-effective time point to introduce therapy to augment the host immune response than when the primary tumor is still present and serving as a more potent antigen. Further, neoadjuvant therapy not only may augment a more robust host immune response but also may serve to sustain this stronger response over the course of treatment [23].

Targeted anti-BRAF and anti-MEK therapy have also been shown to be effective for cytoreduction in BRAF-mutated melanoma. As a significant proportion of melanoma is BRAF-mutated, targeted therapy may be beneficial in the neoadjuvant setting, and in combination with immunotherapy [24]. Targeted therapy may be particularly beneficial in this setting due to their mechanism of action and rapid onset of response. BRAF and MEK are both kinases that are involved in regulating the proliferation and growth of melanoma cells via the mitogen-activated protein kinase/extracellular signal-related kinase (MAPK/ERK) pathway [25,26]. Small molecule inhibitors of these kinases thus prevent disease progression by preventing signal transduction that activates this pathway. This kinase inhibition occurs rapidly, as does its effect, with response seen on the order of days [27,28]. This is a much more rapid response than that seen with checkpoint blockade, as targeted therapy does not require the generation and amplification of an anti-tumor immune response to take effect. Further, in BRAF-mutated melanoma, targeted therapy has been shown to have a high response rate, up to 100% [27,28]. As it is rapid-onset and effective at controlling disease in relevant patient populations, targeted therapy is well-suited as a neoadjuvant treatment. It would be unlikely to delay surgery as its response is seen in days, and it would likely decrease the size and extent of tumors, decreasing the extent of surgery and thus operative morbidity [29,30]. Finally, oncolytic viruses may be leveraged in the neoadjuvant setting. They have been shown to increase the immunogenicity of tumors, and as such, may also be effective in combination with neoadjuvant immunotherapy. This may be of particular importance in patients whose tumors are “cold” [31,32,33].

Finally, neoadjuvant therapy may allow for a more thorough assessment of treatment response and post-treatment recurrence. Neoadjuvant therapy allows for ready comparison of pre-treatment and on-treatment samples, both facilitating clinical decision-making as well as potential future research [34]. Further, the current paradigm of adjuvant treatment does not allow for assessment of treatment response of the primary lesion and only has recurrence versus no recurrence as a post-treatment endpoint. Although the timing and potential combination of neoadjuvant/adjuvant therapy has not yet been defined, a combination approach would allow for more time points to assess treatment response and thus may yield more accurate prognostication of patient course. 

## 3. Neoadjuvant Trials

### 3.1. Trial Endpoints

Herein we highlight and summarize recent landmark neoadjuvant clinical trials in melanoma. While each differs with respect to trial design and drug(s) under investigation, consistency in trial endpoints among the contemporary neoadjuvant trials has allowed for meaningful comparisons between studies. Common endpoints reported among trials have included drug toxicity, radiographic response, and survival (relapse free survival, distant metastasis-free survival, and overall or melanoma-specific survival). However, reporting on pathologic response remains a critical endpoint to neoadjuvant trials, as prior studies have demonstrated this metric to be a significant predictor of outcomes. The International Neoadjuvant Melanoma Consortium (INMC) has proposed standardized criteria when reporting pathologic responses, to ensure consistency across trials [35]. According to the INMC criteria, pathologic complete response (pCR) was defined as the absence of tumor in the treated lesion; near pCR reflected less than or equal to 10% remaining tumor in the treated lesion; partial pathologic response (pPR) was defined as greater than 10% but less than or equal to 50% remaining tumor in the treated lesion; and pathologic no response (pNR) was defined as greater than 50% of the tumor bed occupied by viable tumor cells [36]. In this review, the included trials highlight the importance of these endpoints and how they have helped shape the current neoadjuvant treatment landscape for high-risk melanoma. Studies can be seen in a summarized format in Table 1. 

### 3.2. Immunotherapy

#### 3.2.1. Pembrolizumab in Resectable Stage III–IV Disease

Pembrolizumab, an anti-PD-1 antibody was tested first in the neoadjuvant setting by Huang et al., using a single dose of pembrolizumab (200 mg) followed by complete resection three weeks later in 29 patients with stage III/IV resectable melanoma. The primary endpoints in this trial were safety and immune response. They identified a potent anti-tumor response at three weeks, with 8/27 evaluable patients (30%) experiencing a complete or near pCR (≤10% viable tumor) after a single dose. At the time of the last follow-up, those that had a complete or major pathologic response following a short course of neoadjuvant pembrolizumab remained disease free at 24 months. Additionally, they observed a reinvigoration of CD8 T cells, peaking seven days after therapy. This suggests that PD-1 blockade augments an early innate anti-tumoral T-cell response [37].

#### 3.2.2. Nivolumab vs. Ipilimumab + Nivolumab in Resectable Clinical Stage III or Oligometastatic Stage IV Disease

Amaria et al. performed a clinical trial of neoadjuvant nivolumab (3 mg/kg) versus combined nivolumab (1 mg/kg) and ipilimumab (3 mg/kg) in 23 patients with resectable clinical stage III or oligometastatic stage IV disease. This was followed by surgical resection and adjuvant nivolumab (3 mg/kg) for six months in both groups. The primary endpoints in this trial were clinical responses (including radiologic, pathologic, and survival rates) and comparison of immunologic biomarkers. Treatment with combined ipilimumab and nivolumab yielded overall response rates of 73% and pCR rates of 45% compared to treatment with nivolumab monotherapy which yielded an overall response rate of 25% and pCR rate of 25%. Notably, toxicity rates differed significantly between the two treatment arms, with 73% of patients receiving combination therapy reporting grade 3 adverse events compared to only 8% in the monotherapy arm. Tumor analysis showed higher lymphoid infiltrates in responders to combination therapy, but a more clonal and diverse T cell response in nivolumab monotherapy [38].

#### 3.2.3. OpACIN Trial: Ipilimumab + Nivolumab in Resectable Stage III with Palpable Disease

In the Optimal Adjuvant Combination Scheme of Ipilimumab and Nivolumab in Melanoma Patients (OpACIN) trial, Blank et al. investigated the difference in clinical outcomes between patients receiving ipilimumab (3 mg/kg) in combination with nivolumab (1 mg/kg), a monoclonal antibody against PD-1, in the neoadjuvant vs. adjuvant setting in 20 patients with stage III melanoma with clinically palpable nodal disease. The primary co-endpoints in this trial were safety and feasibility, and a comparison of the immune-activating capacity. In the neoadjuvant arm, 7/9 patients (78%) achieved profound pathologic responses, three of whom had a pCR and no recurrence with a median follow-up of 25.6 months. They found no difference in adverse events between the two groups, though grade 3–4 adverse events were experienced in 9/10 (90%) patients of each arm. Analysis of peripheral blood revealed that neoadjuvant treatment expanded more tumor-resistant T cell clones compared to adjuvant therapy, suggesting neoadjuvant immunotherapy may yield a more potent anti-tumoral immune response compared with adjuvant immunotherapy [39].

#### 3.2.4. OpACIN-Neo Trial: Ipilimumab + Nivolumab—Resectable Stage III 

Rozeman et al. expanded on their previous study to identify an optimal combination dosing schedule of neoadjuvant ipilimumab in combination with nivolumab in 86 patients with resectable stage III melanoma. Patients were assigned to one of three dosing groups: [A] two cycles of ipilimumab 3 mg/kg plus nivolumab 1 mg/kg every three weeks; [B] two cycles of ipilimumab 1 mg/kg plus nivolumab 3 mg/kg every three weeks; [C] two cycles of ipilimumab 3 mg/kg once every three weeks followed by two cycles of nivolumab 3 mg/kg once every two weeks. The primary endpoints in this trial were safety as well as radiological and pathological response. Pathologic responses were observed in 64 (74%) of patients treated. The breakdown for treatment group and pathologic response was as follows: 24 (80%) patients in [A], 23 (77%) in [B], and 17 (65%) in [C]. Of the 30 participants in group [A], 14/30 (47%) had a pCR, 7/30 (23%) had a near pCR, 3/30 (10%) had a pPR, and 6/30 (20%) had a pNR. Of the 30 participants in group [B], 17/30 (57%) had a pCR, 2/30 (7%) had a near pCR, 4/30 (13%) had a pPR, and 7/30 (23%) had a pNR. Of the 26 participants in group [C], 6/26 (23%) had a pCR, 6/26 (23%) had a near pCR, 5/26 (19%) had a pPR, 8/26 (38%) had a pNR, and 1/26 (4%) was not evaluable. At the time of study completion, none of the patients who achieved a pathologic response had relapsed. Additionally, adverse events differed between treatment arms. Grade 3–4 adverse events were reported in 12 (40%) patients in group [A], six (20%) in [B], and 13 (50%) in [C], suggesting the group [B] regimen was best tolerated and resulted in similar response rates as the high ipilimumab dosing [40].

#### 3.2.5. Survival and Biomarker Analysis of OpACIN and OpACIN-Neo Trials 

Aiming to define the durability of pathologic responses in their neoadjuvant immunotherapy trials, Rozeman et al. performed survival and biomarker data analyses from the OpACIN and OpACIN-neo trials. They found that after a median follow-up of four years, none of the patients in the OpACIN study with a pathologic response (n = 7/9) had relapsed, and in the OpACIN-neo trial (n = 86), the 2-year estimated relapse-free survival was 84% for all patients. The relapse-free survival was 97% for patients achieving a pathologic response and 36% for non-responders. In their analysis of biomarker data, a high tumor mutational burden (TMB) and high interferon-gamma (IFN-γ)-related gene expression score was associated with improved pathologic response and reduced recurrence. Both have been shown to correlate with improved checkpoint blockade efficacy [41,42,43]. The pathologic response rate was 100% in patients with a high TMB/high IFN-γ-related gene expression score, compared to a pathologic response rate of only 39% in patients with a low TMB/low IFN-γ-related gene expression score. This study revealed that neoadjuvant ipilimumab plus nivolumab can produce a durable, disease-free interval in a high proportion of patients [44].

#### 3.2.6. PRADO Trial: Response Directed Therapy after Neoadjuvant Ipilimumab and Nivolumab in Stage III Melanoma 

While the OpACIN and OpACIN-neo trials demonstrated that pathologic response was correlated with disease-free survival, Reijers et al. aimed to investigate the utility of using pathologic response after neoadjuvant therapy as a criterion for further treatment personalization in the PRADO extension cohort of the OpACIN-neo trial (Figure 1). This trial included 99 patients with clinical stage IIIB-D melanoma who received neoadjuvant ipilimumab (1 mg/kg) and nivolumab (3 mg/kg). Co-primary endpoints in this trial were pathologic response rate, RFS at 2 years for patients achieving a major pathological response (pCR or near pCR), and RFS at 2 years for patients achieving pNR. Patients who achieved a major pathologic response in their index lymph node did not undergo therapeutic lymph node dissection or adjuvant therapy. In contrast, patients with pPR underwent therapeutic lymph node dissection only, and patients with a pNR underwent therapeutic lymph node dissection and adjuvant therapy. Pathologic responses were observed in 71 of 99 (72%) patients, including 48 (49%) with a pCR and 12 (12%) with a near pCR. As such, a major subset of patients did not undergo therapeutic lymph node dissection, thus reducing surgical morbidity. The 24-month relapse-free survival for patients achieving a major pathologic response (pCR or near pCR) was 93%, with only one patient developing distant metastasis. Of the partial responders who underwent therapeutic lymph node dissection (8/11), the 24-month relapse-free survival was 64%. Non-responders had a 24-month relapse-free survival of 71%. Of the 21 participants with pNR, 7 started adjuvant nivolumab, 10 started adjuvant dabrafenib and trametinib (BRAF/MEK inhibition), and 4 did not receive adjuvant therapy due to treatment-related adverse events. Recurrence was noted in two of the 7 participants at data cutoff who were started on adjuvant nivolumab (RFS at 2 years of 71%), 3 of the 10 participants at data cutoff on dabrafenib and trametinib (RFS at 2 years of 90%), and 2 of the 3 participants at data cutoff who did not receive adjuvant therapy (RFS at 2 years of 33%). The sample sizes of these adjuvant treatment sub-groups were too small to meaningfully compare outcomes. Additionally, 8/21 participants with pNR received adjuvant radiotherapy. Overall, the neoadjuvant ipilimumab plus nivolumab regimen was well tolerated, with grade 3–4 adverse events observed in only 22 (22%) of patients within the first 12 weeks [45].

#### 3.2.7. Relatlimab + Nivolumab in Stage III or Oligometastatic Melanoma

This regimen had previously been shown to be more effective than single-agent therapy with nivolumab in unresectable diseases by the RELATIVITY-047 trial [46]. As such, Amaria et al. explored its utility in resectable disease. In this study, patients received two doses of nivolumab (480 mg)/relatlimab (160 mg) before surgery, followed by ten doses of the same regimen as adjuvant therapy. The primary endpoints in this trial were pCR rate, safety, and efficacy. Seventeen (57%) patients achieved a pCR, and twenty-one patients (70%) in total achieved any pathologic response. Importantly, no patients experienced grade 3 or 4 adverse events. Those with any pathologic response had significantly improved 1- and 2-year survival (100% and 92% vs. 88% and 55%, *p* = 0.005). Pathologic response was also found to be associated with both increased immune cell infiltration at baseline as well as decreased M2 macrophages during treatment. This study suggests that relatlimab/nivolumab is an effective immunotherapy with a more favorable safety profile compared with other regimens [47].

### 3.3. Neoadjuvant vs. Adjuvant Single-Agent Immunotherapy

In the Southwest Oncology Group S1801 trial, Patel et al. compared neoadjuvant and adjuvant pembrolizumab with purely adjuvant pembrolizumab in patients with stage IIIB to IVC melanoma (Figure 2). The combination arm received three doses of neoadjuvant pembrolizumab (200 mg), followed by surgery and 15 adjuvant doses. The purely adjuvant arm received 18 doses of pembrolizumab after surgery. The primary endpoint in this trial was event-free survival (EFS) at two years. The study demonstrated that the neoadjuvant/adjuvant arm had a significantly longer EFS at 2 years, 72%, versus 49% in the adjuvant arm (*p* = 0.004). The two groups did not differ in the rates of serious adverse events (12% vs. 14%), suggesting neoadjuvant/adjuvant therapy was well-tolerated compared with standard-of-care. More importantly, the study directly addressed the question of treatment sequence and provided the first evidence highlighting the superiority of neoadjuvant over adjuvant therapy in preventing relapses [48].

### 3.4. Targeted Therapy

#### 3.4.1. BRAF/MEK Inhibitor Combination 

##### NeoCombi Trial: Dabrafenib + Trametinib—Resectable Clinical Stage IIIB-C Disease 

Long et al. investigated the use of neoadjuvant dabrafenib combined with trametinib in 35 patients with resectable, stage IIIB-C BRAF mutated melanoma. Patients received dabrafenib (150 mg) plus trametinib (2 mg) daily for 12 weeks before surgery, followed by an additional 40 weeks of adjuvant therapy. The primary endpoint in this trial was pCR and RECIST response. With a median follow-up of 27 months, 17 patients (49%) achieved a pCR and 18 (51%) achieved a pPR. At the completion of the study, 20/35 patients (57%) had recurred, 8/17 (47%) from the pCR group, and 12/18 (67%) from the pPR group. Median relapse-free survival was 30.6 months in those with a pCR and 18.0 months in those with a pPR. The treatment was well tolerated with ten patients (29%) experiencing a grade 3–4 adverse event. Tissue analysis demonstrated a higher proliferative index (Ki67-positive melanoma cells) and a pre-existing immune response with CD8-positive T cells in baseline melanoma biopsy samples of patients who had a pCR. While neoadjuvant dabrafenib plus trametinib resulted in high pathologic response rates in comparison to neoadjuvant anti-PD1 immunotherapy, patients with complete pathological responses still experienced a high risk of recurrence [49].

##### Dabrafenib + Trametinib in Resectable Clinical Stage III or Oligometastatic Stage IV Disease

Adjuvant dual BRAF and MEK inhibition has previously been shown to be effective and well-tolerated in stage IV BRAF-mutated melanoma [50]. Amaria et al. investigated the use of neoadjuvant and adjuvant dabrafenib and trametinib vs. adjuvant-only therapy in 21 patients with surgically resectable, clinical stage III and oligometastatic stage IV BRAF-mutated melanoma. Patients assigned to standard of care underwent definitive surgery followed by targeted therapy. Patients in the neoadjuvant plus adjuvant group received eight weeks of neoadjuvant daily dabrafenib (150 mg) and trametinib (2 mg) followed by surgery, and adjuvant therapy on the same regimen for a total of 52 weeks of treatment. The primary endpoint in this trial was EFS at 12 months. Notably, this trial was stopped early after safety analysis revealed significantly longer EFS with neoadjuvant plus adjuvant dabrafenib and trametinib (19.7 months) compared to standard of care (2.9 months). In the neoadjuvant/adjuvant group, 7/12 (58%) patients achieved a pCR with an additional 2/12 (17%) patients having a pPR. The neoadjuvant/adjuvant arm was generally well tolerated, with no grade 4 adverse events and few grade 3 adverse events. Molecular and immune profiling identified that patients achieving a pathologic complete response had significantly lower to no detectable baseline expression of phospho-ERK (pERK) in tumor tissue. Further, tumors from patients with a pCR showed little to no remodeling of the T-cell population between baseline and surgery with neoadjuvant therapy, compared to a greater variation in those who did not achieve a pCR. This finding suggests that effector T-cells that contribute to the anti-tumor response may be present before treatment with neoadjuvant dabrafenib and trametinib [51].

**Table 1 cancers-15-03344-t001:** Summary of trials of neoadjuvant therapy in resectable advanced stage melanoma.

Treatment Group	Trial (Registry Number)	Population	Design	Intervention	Primary Endpoint(s)	Response	Toxicity	Findings in Context
Mono-tx anti-PD1	Huang 2019 [37] (NCT02434354)	Resectable stage III/IV	Phase Ib (N = 29)	Neoadj pembro 200 mg × 1, then adj pembro (1 year)	Safety	8/27 (30%) with complete or major path response	No unexpected AEs	PD-1 blockade ↑ anti-tumor T cell response
Comb-tx anti-CTLA4 + anti-PD1	Amaria 2018 [38] (NCT02519322)	Resectable stage III/IV	Phase II (N = 23)	[A] Neoadj nivo 3 mg/kg × 4 [B] Neoadj ipi 3 mg/kg + nivo 1 mg/kg × 3	pCR	pCR in [A] 3/12 (25%), [B] 5/11 (45%)	Grade 3 AEs in [A] 8%, [B] 73%; no grade 4–5 AEs observed	Improved pCR in ipi + nivo comb tx but considerable toxicity
Comb-tx anti-CTLA4 + anti-PD1	OpACIN, Blank 2018 [39] (NCT02437279)	Palpable stage III	Phase Ib (N = 20)	[A] Neoadj ipi 3 mg/kg + nivo 1 mg/kg × 2, then adj ipi + nivo × 2 [B] Adj-only ipi 3 mg/kg + nivo 1 mg/kg × 4	Safety, immune response	Path response in [A] 7/9 (78%), favorable T cell response in [A]	Grade 3/4 AEs in 9/10 pts in each arm	Possible superiority of neoadj tx, but high toxicity
Comb-tx anti-CTLA4 + anti-PD1	OpACIN-neo, Rozeman 2019 [40] (NCT02977052)	Resectable stage III	Phase II (N = 86)	[A] Neoadj ipi 3 mg/kg + nivo 1 mg/kg × 2 [B] Neoadj ipi 1 mg/kg + nivo 3 mg/kg × 2 [C] Neoadj ipi 3 mg/kg × 2 then nivo 3 mg/kg × 2	Safety, rads/path response	Rads objective response—[A] 63%, [B] 57%, and [C] 35%; path response—[A] 80%, [B] 77%, and [C] 65%	AEs grade 3/4 in [A] 40%, [B] 20%, and [C] 50%	High path response rate in better tolerated dosing sch [B] of neoadj ipi + nivo
Comb-tx anti-CTLA4 + anti-PD1	PRADO (OpACIN-neo expansion cohort), Reijers 2022 [45] (NCT02977052)	Nodal stage IIIB–D	Phase II (N = 99)	Neoadj ipi 1 mg/kg + nivo 3 mg/kg × 2, then assess ILN MPR in ILN -> TLND and adj tx omitted pPR in ILN -> TLND onlypNR in ILN -> TLND and adj chemo + radiotherapy	Safety, pRR, RFS	ILN resected 90/94 pts 1st attempt; MPR 61%, pPR 11%, pNR 21%; TLND omitted in 59/60 pts with MPR; 24-mo RFS 93% in MPR, 64% pPR, and 71% pNR	Grade 3/4 AEs in 22%	Supports response-driven personalization of tx after neoadj ipi+nivo
Comb-tx anti-PD1 + anti-LAG3	Amaria 2022 [47] (NCT02519322)	Resectable stage III/IV	Phase II (N = 30)	Neoadj nivo 480 mg + relatlimab 180 mg × 2, then adj nivo + relatlimab × 10	Safety, pCR	pCR in 57%, near pCR 7%, pPR 7%, and pNR 27%; 2-year RFS 91% in pCR, 92% any path response, and 55% without path response	No grade 3–4 AEs in neoadj, 26% grade 3–4 AEs in adj	Comparable pCR and safety profile to other neoadj combination therapies
Mono-tx anti-PD1	SWOG 1801, Patel 2023 [48] (NCT03698019)	Resectable stage IIIB–IVC	Phase II (N = 313)	[A] Neoadj pembro 200 mg × 3, then adj pembro × 15 [B] Adj-only pembro × 18	Event-free survival	EFS at 2 years in [A] 72%, and [B] 49%	AEs grade 3+ in [A] 12%, and [B] 14%	EFS significantly longer in neoadj arm
Comb-tx BRAFi + MEKi	NeoCombi, Long 2019 [49] (NCT01972347)	Resectable BRAF-mutated stage IIIB–C	Phase II (N = 35)	Neoadj dabrafenib 150 mg BID + trametinib 2 mg daily × 12 wks, then adj dabrafenib + trametinib × 40 wks	pCR, RECIST response at 12 wks	35/35 had a path response, 17/35 (49%) had a pCR; RECIST response in 30/35 (86%), complete in 16/35 (46%), partial in 14/35 (40%)	Grade 3–4 AEs in 10/35 (29%)	Well tolerated and effective comb targ neoadj tx in resectable stage III BRAF-mutated melanoma
Comb-tx BRAFi + MEKi	Amaria 2018 [51] (NCT02231775)	Resectable BRAF-mutated stage III/IV	Phase II (N = 21)	[A] Neoadj dabrafenib 150 mg BID + trametinib 2 mg daily × 8 wks, then adj dabrafenib + trametinib × 44 wks [B] Adj standard-of-care	EFS at 12 mos	Median EFS in [A] 19.7 months, and [B] 2.9 months; pCR in [A] 7/12 (58%)	No grade 4 AEs in [A]	Support rationale for comb targ neoadj tx in resectable stage III/IV BRAF-mutated melanoma

Abbreviations: mono-tx—mono-therapy, comb-tx—combination therapy, neoadj—neoadjuvant, adj—adjuvant, ipi—ipilimumab, IFNa—interferon alpha, pembro—pembrolizumab, nivo—nivolumab, ILN—index lymph node, TLND—therapeutic lymph node dissection, pNR—pathologic non-response (>50% viable tumor), pPR—pathologic partial response (>10 to <50% viable tumor), MPR—major pathologic response (<10% viable tumor), near pCR—<10% viable tumor, pCR—absence of viable tumor, pRR—pathologic response rate, wks—weeks, BID—twice daily, RFS—relapse-free survival, EFS—event-free survival, AEs—adverse events, sch—schedule, targ—targeted.

## 4. Discussion

The current literature supports a role for neoadjuvant immunotherapy in the treatment of advanced-stage resectable melanoma. Available studies demonstrate that neoadjuvant therapy is largely associated with improved relapse-free survival, may reduce the extent of required surgery, and is largely well-tolerated. Based on biomarker data, the efficacy is likely due to the mechanisms described above—neoadjuvant immunotherapy enhances host anti-tumor response. The result is larger populations of effector T-cells and a more sustained immune response compared with adjuvant therapy. Neoadjuvant targeted therapy, while mechanistically different, has also proven to be effective, resulting in significant cytoreductive potential and allowing for a cohort of patients that were initially deemed unresectable to become surgical candidates. Finally, neoadjuvant treatment allows for assessment of pre-operative treatment response, valuable pathologic and biomarker data, and even tailoring of future treatment based on the neoadjuvant course.

The efficacy of neoadjuvant therapy as shown by the described trials also yields potential considerations for future research. First, although effective generally with currently available therapies, elucidating the optimal treatment combinations, timing, and duration of treatment is of great interest. One current study investigates these questions further and utilizes pathologic response to neoadjuvant therapy to direct future treatments in patients with stage III melanoma (NCT04013854) [52]. According to the trial design, patients receive one dose of nivolumab and undergo surgery. Those who have a near or pCR receive adjuvant nivolumab, and those who do not are randomized to receive nivolumab alone or nivolumab (1 mg/kg) plus ipilimumab (3 mg/kg). Similar to the PRADO trial, this study highlights how pathologic response afforded through a neoadjuvant approach enables tailored treatment strategies to be studied. As the uptake and use of neoadjuvant therapy become more common, there will be more data available for study and subsequent evidence-based guidance of treatment timing and course. Along with this trial, ongoing and future trials of neoadjuvant therapy in resectable late-stage melanoma are described in Table 2.

Finally, although current immunotherapy is effective, emerging treatments in the neoadjuvant setting or novel immunotherapies may prove to have an additive benefit in advanced-stage melanoma treatment. As an example, immune checkpoint blockade is subject to the development of resistance by the primary tumor by way of T cell exhaustion and alteration of the tumor microenvironment to a more immunosuppressive phenotype. As such, immunotherapy directed towards damage- or pathogen-associated molecular patterns (DAMPs/PAMPs), toll-like receptors (TLRs), and other immunoregulatory domains such as Tcell immunoreceptors with Ig and ITIM domains (TIGIT) may further enhance anti-tumor immunity and could prove effective in patients whose disease is resistant to conventional immune checkpoint blockade [53]. TIGIT is a recently identified receptor whose putative function is thought to be promoting an immunosuppressive dendritic cell phenotype [54,55]. One ongoing study investigates the efficacy of tiragolumab (anti-TIGIT) in combination with atezolizumab (PD-L1 blockade) in BRAF wild-type stage III melanoma and with both atezolizumab and cobimetinib (MEK-inhibitor) in BRAF mutated stage III melanoma (NCT03554083) as seen in Table 2 [56]. This highlights the potential for further advancement in identifying synergistic immunotherapy/targeted therapy regimens. Further, novel therapy leveraging the cyclic GMP-AMP synthase/stimulator of interferon genes (cGAS/STING) signaling pathway, inhibiting the tumor-growth receptor CD73, or enhancing pyroptosis of melanoma cells through gasdermin activation are all potentially promising targets of immunotherapy in melanoma [57,58,59,60,61,62,63].

## 5. Limitations and Challenges

Despite the promise of neoadjuvant therapy, there still exist limitations in the current literature. A major challenge is the lack of long-term follow-up, both with respect to survival and recurrence after neoadjuvant immunotherapy and targeted therapy. However, based on the INMC pooled analysis, it does appear that neoadjuvant immunotherapy results in a more durable relapse-free interval than targeted therapy [64]. When extrapolating from the metastatic melanoma literature, these observations, supporting neoadjuvant immunotherapy over targeted therapy, were further corroborated, in the DREAMseq trial, which investigated which initial treatment for BRAF-mutated melanoma was more effective [65]. As such, although the data appears to favor immunotherapy, long-term results have yet to be published; however, the 5-year follow-up results from the initial wave of early neoadjuvant trials are likely to be reported soon. The current studies also include a relatively small sample size. In aggregate, the evidence is strong, however individual studies remain limited in trial size.

There are also ongoing challenges to the uptake of neoadjuvant therapy. As mentioned, optimal neoadjuvant treatment schedules and regimens have not been established. From what we can glean from available data, the SWOG 1801 trial was the first to demonstrate that treatment sequence matters. This trial has become central to the paradigm shift towards neoadjuvant therapy in melanoma as it provides strong evidence that neoadjuvant therapy is more effective than adjuvant therapy alone. However, as newer combination approaches emerge, which result in higher pathologic response rates, the use of single-agent checkpoint blockade in the neoadjuvant setting will likely have less of a role. Secondly, pathologic response appears to be one of the most important reasons for a neoadjuvant approach. When compared to radiographic response, which remains a common study endpoint in advanced/metastatic trials, pathologic response has arisen as a stronger prognostic factor. Radiographic changes after a short course of neoadjuvant therapy is often delayed, making this metric less reliable. As a result, pathologic response is now being used to shape trial designs (i.e., PRADO, NCT04013854) and is one example of how cancer care is evolving towards a more personalized approach. Next, while much attention has been placed on using pathologic response to refine neoadjuvant strategies, it exposed that adjuvant therapy will continue to have a role, especially among patients that derive anything less than a pCR or near pCR. Follow-up data from the PRADO trial demonstrated that omission of adjuvant therapy among patients with a pPR resulted in worse outcomes when compared to the pNR cohort that underwent both surgery and adjuvant therapy. Important questions that arose from the PRADO trial include which type of adjuvant therapy is better, targeted versus immunotherapy among non-responders (pPR/NR), along with how to manage patients that had a pCR, but subsequently relapsed. As such, the timing of treatment, dosages, and regimens must continue to be studied in order to shift neoadjuvant therapy from investigational to guideline-based.

Lastly, another challenge to the adoption of a neoadjuvant approach is the potential that patients who would have been otherwise cured by upfront surgery, are placed on treatment regimens that are costly with potential toxicities. Initial concerns that a neoadjuvant approach would lead to a “missed opportunity” for a curative procedure due to disease progression have not been observed. Instead, identifying patients, perhaps through new biomarkers analysis (i.e., TMB/IFN-γ profile), who would benefit most from a neoadjuvant approach remains of critical importance to avoid the potentially irreversible toxicities of these costly treatments.

## 6. Conclusions

The mainstay of treatment in melanoma is initial surgery followed by adjuvant treatment for resectable late-stage disease. However, the literature and clinical trials support neoadjuvant immunotherapy as not only non-inferior but a potentially more efficacious treatment, challenging the current paradigm. Neoadjuvant immunotherapy has been shown to yield a more robust anti-tumor response by augmenting an existing patient immune response. This immune response elicited by neoadjuvant therapy has also been shown to be sustained until and beyond surgery and has been associated with improved recurrence-free survival. Additionally, the neoadjuvant approach allows for the assessment of treatment efficacy both clinically and in terms of pathologic response, which is beneficial both for prognostication as well as tailoring future treatment. Although seemingly effective and an improvement on current treatment guidelines, several salient questions remain. First, although broadly effective, the optimal neoadjuvant treatment timing and schedule remain to be elicited. Further, existing immunotherapy has limitations that may be solved through novel combination strategies, completely novel immunotherapies, and continued large-scale trials with longer follow-up. At this time, many questions remain surrounding neoadjuvant therapy, which limits its widespread adoption in clinical practice. Instead, trial enrollment when feasible, should remain a priority. Regardless, neoadjuvant immunotherapy has transformed the treatment landscape for resectable advanced staged melanoma and will continue to have a major role.

## Figures and Tables

**Figure 1 cancers-15-03344-f001:**
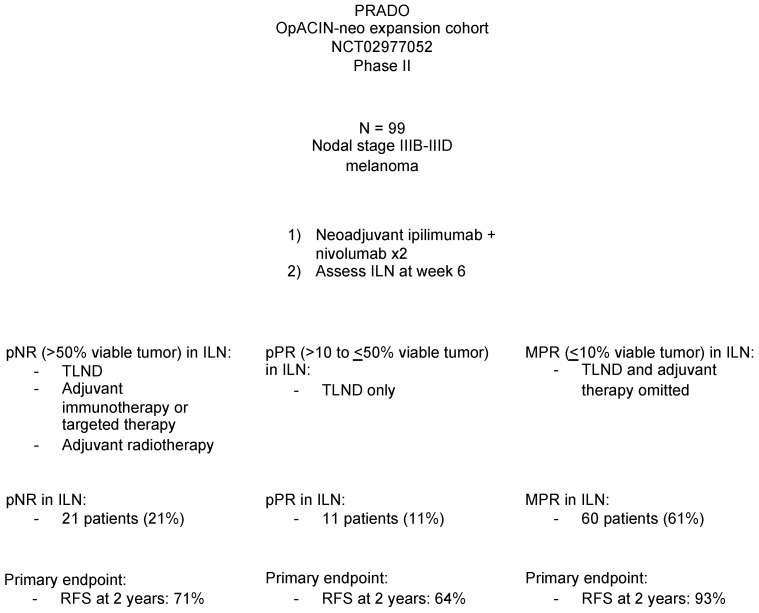
Findings of the PRADO trial. Abbreviations: ILN—index lymph node, TLND—therapeutic lymph node dissection, pNR—pathologic non-response (>50% viable tumor), pPR—pathologic partial response (>10 to <50% viable tumor), MPR—major pathologic response (<10% viable tumor), RFS—relapse-free survival.

**Figure 2 cancers-15-03344-f002:**
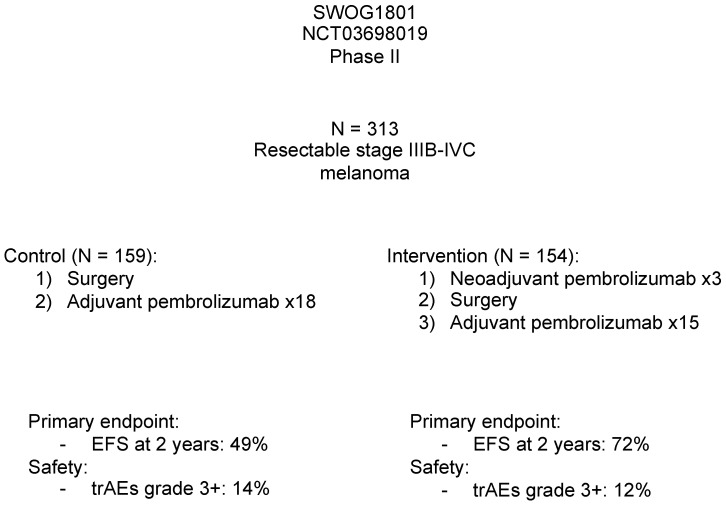
Findings of the SWOG1801 Trial. Abbreviations: EFS—event-free survival, trAEs—treatment-related adverse events.

**Table 2 cancers-15-03344-t002:** Summary of Ongoing and Future Trials of Neoadjuvant Therapy in Resectable Stage III/IV Melanoma.

Treatment Group	Trial (Registry Number)	Population	Design	Intervention	Primary Endpoint(s)
Comb-tx Anti-CTLA4 + anti-PD1	NCT04013854	Resectable stage III melanoma	Phase II (N = 60)	[A] One dose of nivo IV; Surgery; If pCR, nivo IV for up to 1 year [B] One dose of nivo IV; Surgery; If <near pCR, nivo IV for up to 1 year [C] One dose of nivo IV; Surgery; If <near pCR, ipi IV + nivo IV for 4 doses, then nivo IV alone for a total of 1 yr	RFS
Comb-tx Anti-CTLA4 + anti-PD1	NADINA NCT04949113	Macroscopic stage III melanoma	Phase III (N = 420)	[A] Ipi IV + nivo IV q3 wks, 2 cycles; TLND; Nivo IV q4 wks, up to 11 cycles IF pPR or pNR If BRAF V600E/K mutant, dabrafenib + trametinib for 46 wks instead of nivo [B] TLND; Nivo q4 wks, 12 cycles	EFS
Comb-tx T-VEC + anti-PD1	NCT04330430	Stage III–IV melanoma	Phase II (N = 24)	T-VEC intra-lesional inj × 4 q2 wks after 3 wks of 1st dose; nivo IV × 3 q2 wks starting after 2nd T-VEC course	pCR
Comb-tx T-VEC + anti-PD1	NCT03842943	Resectable stage III cutaneous melanoma	Phase II (N = 28)	T-VEC intra-lesional inj q3 wks, up to 6 mos; pembro IV q3 wks, up to 6 mos, then q3 wks for 1 year in adj setting	pCR
Comb-tx Anti-PD1 + hGM-CSF HSV	NCT04197882	Resectable stage III and IV (M1a) melanoma	Phase Ib (N = 33)	Toripalimab IV q2 wks × 6, OrienX010 intratumoral inj q2 wks × 6; Surgery; Toripalimab IV q3 wks for up to 1 yr	pCR, RECIST response
Comb-tx Anti-PD1 + BRAFi + MEKi	Neo Trio NCT02858921	Resectable BRAF V600 mutant stage IIIB/IIIC melanoma	Phase II (N = 60)	[SEQ] Dabrafenib PO BID + trametinib PO QD × 1 wk, then followed by pembro IV at wks 1, 3, and 6, then q3 wks from wks 6–36 [CON] Dabrafenib PO BID + trametinib PO QD + pembro IV q3 wks for 6 wks; then pembro alone for 46 wks [ALONE] Pembro IV q3 wks for 52 wks	pCR
Comb-tx Anti-PDL1 + BRAFi + MEKi, Anti-PDL1 + anti-TIGIT	NeoACTIVATE NCT03554083	Stage III cutaneous melanoma	Phase II (N = 30)	[A] Vemurafenib PO BID on days 1–28, cobimetinib PO QD on days 1–21, atezolizumab IV on days 1 and 15 of cycles 2 and 3, up to 3 cycles; Surgery; Atezolizumab IV on day 1, repeats q3 wks, up to 8 cycles [B] Cobimetinib as in arm [A], atezolizumab IV on days 1 and 15, repeats q4 wks up to 3 cycles; Surgery; Atezolizumab on day 1, repeats q3 wks, up to 8 cycles [C] Atezolizumab IV on day 1, tiragolumab IV on day 1, repeats q3 wks, up to 4 cycles	pCR, RFS
Comb-tx Anti-CTLA4 + anti-PD1, Anti-PD1/LAG3, Anti-PDL1 + anti-TIGIT, Anti-PD1/LAG3 + anti-TIGIT	Morpheus-Melanoma NCT05116202	Resectable stage III (cohort 1) and stage IV (cohort 2) melanoma	Phase Ib/II (N = 191)	[A1] nivo IV on day 1, ipi IV on day 1, repeat q3 wks, 2 cycles [B1] RO7247669 IV on day 1, repeat q3 wks, 2 cycles [C1] Atezolizumab IV on day 1, tiragolumab on day 1, repeat q3 wks, 2 cycles [D1] RO7247669 IV on day 1, tiragolumab on day 1, repeat q3 wks, 2 cycles [A2] RO7247669 IV on day 1, tiragolumab on day 1, repeat q3 wks, until unacceptable toxicity or loss of clinical benefit	pCR, RECIST response
Comb-tx Anti-PD1 + multiple RTKi	Neo PeLe NCT04207086	Resectable stage III melanoma	Phase II (N = 40)	Pembro + lenvatinib for 6 wks; Surgery; Pembro for 46 wks	pCR, immune response
Comb-tx Anti-PD1 + multiple RTKi	Neo PeLeMM NCT05545969	Resectable mucosal melanoma	Phase II (N = 44)	Pembro + lenvatinib for 6 wks, Surgery, Pembro alone for 46 wks	pCR, immune response
Comb-tx Anti-PD1 + multiple RTKi	NCT04622566	Resectable mucosal melanoma	Phase II (N = 26)	Pembro IV on day 1, repeat q3 wks, for 6 wks, lenvatinib PO QD for 6 wks; Surgery; Pembro IV on day 1, repeat q3 wks, up to 15 cycles	pCR

Abbreviations: comb-tx—combination therapy, T-VEC—talimogene laherparepvec, hGM-CSF HSV—human cytokine granulocyte-macrophage colony-stimulating factor herpes simplex virus, TIGIT—T cell immunoreceptor with Ig and ITIM domains, LAG3—lymphocyte-activation gene 3, RTKi—receptor tyrosine kinase inhibitor, nivo—nivolumab, ipi—ipilimumab, pembro—pembrolizumab, TLND—therapeutic lymph node dissection, pCR—absence of viable tumor, near pCR—≤10% viable tumor, pPR—pathologic partial response (>10 to ≤50% viable tumor), pNR—pathologic non-response (>50% viable tumor), IV—intravenous, PO—per oral, BID—twice daily, QD—daily, inj—injection, q—every, yr—year, mos—months, wks—weeks, adj—adjuvant, RFS—relapse-free survival, EFS—event-free survival.

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
