# Peer review of "The Current State of Neoadjuvant Therapy in Resectable Advanced Stage Melanoma"

_cancers, 2023, doi:10.3390/cancers15133344_

Round 1

Reviewer 1 Report

This is a really useful and timely review in neoadjuvant therapies in melanoma. My comments are general, as the manuscript is well-written and structured. 

1) Although covering all potential trials with neoadjuvant treatment may be helpful, in this instance I would suggest that the small initial triuals (IPI in IIIB-IV) and Pembro in stage III-IV dont need to be highlighted with paragraphs - I would suggest including all potential data in Table 1 -  but keep the review focussed on large recent trials - since there are many. This also provides space for a little more discussion and clarification - possibly even some figures of some of the most important clinical trial data (PRADO for instance_

2) defining pathological response is important here - as new definitions, near complete, major, complete, partial etc are used in the neoadjuvant setting - and throughout the manuscript the authors use some of these without clear definitions. This is also worthy of discussions, how where these defined, is there consistency in the literature - and there has been studies confirming reproducibility of pathologic assessments (these references and details should be included).

3) Perhaps some discussion on the types of adjuvant therapy used in some of these trials - for instance in PRADO some patients were treated with combiDT, others with IO - although the numbers are small, was this impactful to patient outcomes

4) Some discussion on the implications of treating patients who may be cured post surgery with expensive and potentially toxic drugs should be mentioned. This is particularly pertinent as we move into the stage II adjuvant setting. The final concluding sentence - that neoadjuvant therapy has a role is also a  little weak. Clearly this therapy is effective, but it would be valuable to have the authors thoughts on neoadjuvant vs adjuvant, neoadjuvant targeted therapy vs immunotherapy. These are the big questions clinically and more discussion should be included.

Fine

Reviewer 2 Report

In the current review, Bushara O et al. summarize the rationale and the results of recent clinical trials in the neoadjuvant treatment in advanced melanoma. Overall, the review is a summary of recent results from clinical trials - future recommendations and challenges, which would be interesting to read in a review of this hot topic, are not mentioned in the manuscript. The authors mention that the "goal of this article is to review clinical trials of neoadjuvant therapy" and indeed, I believe that the current manuscript lacks a critical review of the available data. I have the following recommendations:

- In the introduction section, the authors should mention that the neoadjuvant treatment (tx) should be solely considered in pts with resectable melanoma (not only "advanced"). Also, "downstage" of the tumor is rather unlikely - do the authors mean "size reduction" or "tumor shrinkage"?

- Rationale: the described rationale for the use of the neoadjuvant treatment mostly refers to immunotherapy. Any proposed pathomechanism for the use of BRAF/MEK inhibitors in the neoadjuvant setting? Any evidence on why this could be superior to the adjuvant treatment (if any)?

- Clinical trials: I believe that the SWOG 1801 study should be described separately, as this landmark study proved the superiority of the neoadjuvant tx to the adjuvant tx. I would probably recommend a new section of adjuvant vs neoadjuvant. 

- The authors should add a table of ongoing studies in the neoadjuvant setting incl. NCT number. 

- Data on patients with partial or complete pathologic response that recur should be mentioned in the clinical trial results, as these include a future challenge (see related Poster from the NKI group at ESMO 2022). 

- Absent of long term follow-up and OS data should be discussed critically. Other challenges include small number of patients and maturity of the pathological criteria. It would be probably interesting to discuss whether the neoadjuvant treatment is ready to be implemented in clinic - challenges to consider?

- Please describe the clinical trial results according to the primary and secondary endpoints, as these influence the clinical trial design.

Moderate editing is required - English language is fine. 

Round 2

Reviewer 2 Report

The authors have adequately addressed my comments. 

-